# Effects of Extreme Light Cycle and Density on Melatonin, Appetite, and Energy Metabolism of the Soft-Shelled Turtle (*Pelodiscus sinensis*)

**DOI:** 10.3390/biology11070965

**Published:** 2022-06-26

**Authors:** Zhonghua Tang, Shifan Liu, Cuijuan Niu

**Affiliations:** Ministry of Education Key Laboratory for Biodiversity Science and Ecological Engineering, College of Life Sciences, Beijing Normal University, Beijing 100875, China; zhonghuatang@cqnu.edu.cn (Z.T.); 201821200073@mail.bnu.edu.cn (S.L.)

**Keywords:** soft-shelled turtle, constant light, constant darkness, density, melatonin, appetite, energy metabolism

## Abstract

**Simple Summary:**

Light is an important factor that affects a variety of physiological functions of animals. Melatonin is known as the hormonal mediator of photoperiodic information to the central nervous system in vertebrates and plays a key role in food intake and energy balance regulation. The present work aimed to examine the effect of extreme light cycles (herein meaning constant darkness and constant light) that animals may encounter under dark culture situations or light pollution on melatonin secretion and melatonin-modulated appetite and energy metabolism of the soft-shelled turtle *Pelodiscus sinensis*. We measured melatonin level, appetite, and energy metabolism-related parameters under various photoperiods and stocking densities during daytime and nighttime. The results demonstrated that (1) melatonin secretion of the turtle was not only affected by light, but also likely to be regulated by unknown endogenous factors and density; and (2) altered plasma melatonin induced by constant darkness and density seemed to be involved in the modulation of energy metabolism (elevated thyroid activity and standard metabolic rate) rather than appetite (unchanged leptin and ghrelin contents, kisspeptin 1, cocaine amphetamine-regulated transcript, and neuropeptide Y mRNA) in Chinese soft-shelled turtles. Our results shed light on the physiological responses of freshwater turtles to extreme light cycles and also contribute important data for turtle culture.

**Abstract:**

Constant darkness and constant light exposure often disturb the circadian rhythm in the behavior and energy metabolism of vertebrates. Melatonin is known as the hormonal mediator of photoperiodic information to the central nervous system and plays a key role in food intake and energy balance regulation in vertebrates. The popularly cultured soft-shelled turtle *Pelodiscus sinensis* has been reported to grow better under constant darkness; however, the underlying physiological mechanism by which darkness benefits turtle growth is not clear yet. We hypothesized that increased melatonin levels induced by darkness would increase appetite and energy metabolism and thus promote growth in *P. sinensis*. In addition, in order to elucidate the interaction of photoperiod and density, juvenile turtles were treated under three photoperiods (light/dark cycle: 24L:0D, 12L:12D, 0L:24D, light density 900 lux) and two stocking densities (high density: 38.10 ind./m^2^, low density: 6.35 ind./m^2^) for 4 weeks, and then the blood and brain tissues of turtles were collected during the day (11:00–13:00) and at night (23:00–1:00) after 2 days of fasting. We examined changes in plasma melatonin levels, food intake (FI), and appetite-related hormones (plasma ghrelin and leptin), as well as growth and energy metabolism parameters such as specific growth rate (SGR), standard metabolic rate (SMR), plasma growth hormone (GH), and thyroid hormone/enzyme activity (plasma triiodothyronine T_3_, thyroxine T_4_, and T_4_5′-deiodinase activity). Moreover, we also assessed the responses of mRNA expression levels of food intake-related genes (kisspeptin 1 (*Kiss1*); cocaine amphetamine-regulated transcript (*CART*); neuropeptide Y (*NPY*)) in the brain. The results showed that under high density, SGR was the lowest in 24L:0D and the highest in 0L:24D. FI was the highest in 0L:24D regardless of density. Plasma melatonin was the highest in 0L:24D under high density at night. SMR increased with decreasing light time regardless of density. Most expressions of the measured appetite-related genes (*Kiss1*, *CART*, and *NPY*) were not affected by photoperiod, nor were the related hormone levels, such as plasma leptin, ghrelin, and GH. However, thyroid hormones were clearly affected by photoperiod. T_3_ level in 0L:24D under high density during the day was the highest among all treatment groups. T_4_ in 24L:0D under high density during the day and T_4_5′-deiodinase activity in 24L:0D under low density at night were significantly reduced compared with the control. Furthermore, the energy metabolism-related hormone levels were higher under higher density, especially during the day. Together, melatonin secretion is not only modulated by light but also likely to be regulated by unknown endogenous factors and density. Altered plasma melatonin induced by constant darkness and density seems to be involved in the modulation of energy metabolism rather than appetite in the soft-shelled turtle.

## 1. Introduction

Light plays critical roles in the growth, development, and other physiological processes of animals. Photoperiod is one of the most crucial signal factors that regulate the proper phasing of behavioral and physiological variables such as feeding, energy metabolism, and neuroendocrine functions [1]. Nowadays, in terms of light control for culture purposes or environmental problems, how animals will respond to extreme light cycles (herein meaning constant darkness or constant light) has received great attention [2,3,4], with the focus mostly on birds and rodents. The soft-shelled turtle *Pelodiscus sinensis* is a freshwater reptile with great economic value, living at the bottom of water with dim light with a preference to feed in the dark [5]. It has been reported that 24 h of darkness is beneficial to the feeding and growth of the soft-shelled turtle [3,5]; however, the underlying physiological mechanism is not yet clear. Few studies have explored the effect of light on physiological processes in turtles.

Melatonin, which is mainly produced in the pineal gland and the retina, is well known to be the hormonal mediator of photoperiodic information to the central nervous system in vertebrates [6,7]. Due to its amphipathic characteristics, melatonin is rapidly released into the blood after synthesis without storage, and thus, plasma melatonin level is often used to reflect melatonin secretion of the pineal gland. Generally, melatonin secretion occurs during the dark phase and is suppressed by light, regardless of the habitat of animals [8]. Interestingly, published results differ in whether melatonin secretion is also regulated by endogenous factors [9,10,11,12,13]. In addition, numerous studies have demonstrated that melatonin exerts profound biological functions, such as food intake [7,14], energy balance regulation [15,16], sleeping regulation [17], antioxidant defense [18,19], immune system activity [20], and reproduction [21]. Thus, we speculate that increased melatonin levels induced by darkness increase appetite and energy metabolism and thus promote growth in *P. sinensis*.

Body weight is determined by the balance between energy intake (feeding) and energy expenditure (such as the energy cost of metabolism and activity). Feeding (appetite) is mainly regulated by neurons in the arcuate nucleus (ARC), that is, NPY/AgRP neurons with the co-expression of neuropeptide Y (NPY) and agouti-related peptide (AgRP), and POMC/CART neurons with co-expression of proopiomelanocortin (POMC) and cocaine-amphetamine-regulated transcript (CART) [22,23,24]. In addition, kisspeptin neurons project to the ARC and exert an effect of reducing food intake by directly stimulating POMC/CART and indirectly suppressing NPY/AgRP neurons [25,26]. Furthermore, NPY/AgRP and POMC/CART neurons of the ARC have been implicated in regulating leptin, ghrelin, and growth hormone effects on energy homeostasis [27,28]. Interestingly, it has been revealed that melatonin affects body weight and energy homeostasis in relation to the above ARC neuron biological process. For instance, incubation of the hypothalamus with melatonin upregulated *NPY* mRNA levels, which stimulated appetite and downregulated *POMC* and *CART* mRNA levels with anorectic effects [29]. Chronic melatonin administration (10 days) reduced specific growth rate (SGR) and food intake, and the expression of leptin mRNA was upregulated, while orexigenic signals such as ghrelin and *NPY* mRNA were downregulated in goldfish (*Carassius auratus*) and zebrafish (*Danio rerio*) [7,14]. The majority of studies showed that higher melatonin reduced the body weight of animals [7,14]. However, in some species, the results were conflicting, such as melatonin administration increased body mass and food intake but decreased plasma thyroxine (T_4_) levels in gray mouse lemurs (*Microcebus murinus*) [30]. External melatonin increased ghrelin concentrations, decreased and subsequently raised growth hormone (GH) levels, and increased body mass indices in raccoon dogs (*Nyctereutes procyonoides*) [31]. Therefore, the responses of growth to melatonin changes are species-specific. Based on the close relationship between melatonin and feeding, as well as energy metabolism, we hypothesized that altered melatonin levels induced by constant darkness will increase appetite and energy metabolism, and thus growth, in the Chinese soft-shelled turtle.

High stocking density is universal in aquaculture and usually elicits adverse effects on cultured animals, such as poor food utilization, depression of growth and food intake, and high mortality [32,33]. It has been reported that the survival rate, SGR, and food intake of Chinese soft-shelled turtle hatchlings decrease with increasing stocking density [34]. In juvenile *P. sinensis*, SGR also decreased with increasing stocking density accompanied by higher stress responses [35]. Thus, it will be interesting to know if the benefits acquired from constant darkness can alleviate the adverse effects of high stocking density in turtle culture.

The present study aimed to investigate the physiological mechanism underlying the effects of extreme light cycles and stocking density on the turtle, from a melatonin regulation perspective. Here, we examined changes in melatonin levels, food intake, and appetite-related hormones (plasma ghrelin and leptin), as well as the growth and energy metabolism parameters SGR, standard metabolic rate (SMR), GH, and thyroid hormone/enzyme activity (triiodothyronine T_3_, T_4_, and T_4_5′-deiodinase activity), under different photoperiod and density treatments sampled during daytime and nighttime, respectively. Moreover, we also assessed the responses of mRNA expression levels of food intake-related genes (*Kiss1*, *CART*, and *NPY*) in the brain. This study will shed light on the physiological response to light of freshwater turtles and also provide useful data for aquaculture of the soft-shelled turtle.

## 2. Materials and Methods

### 2.1. Experimental Animals and Acclimation

Juvenile turtles were acquired from the Baiyun Lake Chinese soft-shelled turtle hatchery (Jinan, Shandong, China) in March 2021. The turtles were housed in tanks (length × width × height: 70 cm × 45 cm × 35 cm) with 6 turtles in each tank. The turtles were maintained under 29 ± 1 °C water temperature, approximately 900 lux light density, and a 12L:12D light/dark cycle (light on: 6:00 and light off: 18:00) for more than 30 days. The photophase (6:00–18:00) and scotophase (18:00–6:00) were called ‘daytime’ and ‘nighttime’, respectively. During acclimation, the turtles were fed to satiation with a commercial powder diet at 12:00 p.m. every day. Uneaten food and feces were removed after one hour of feeding, and approximately half of the water was refreshed in the tank each day.

### 2.2. Experimental Protocol

After acclimation, all turtles were fasted for 2 days, and 108 individuals (118.57 ± 14.30 g) of similar size were selected, weighed, and randomly divided into 6 groups with two stocking density levels (high density: 38.10 ind./m^2^ and low density: 6.35 ind./m^2^) and three photoperiods (light/dark: 24L:0D, 12L:12D and 0L:24D), that is, high density and 24L:0D group, high density and 12L:12D group, high density and 0L:24D group, low density and 24L:0D group, low density and 12L:12D group, and low density and 0L:24D group. The high-density groups in each photoperiod included 2 replicates with 12 turtles per replicate, and the low-density groups in each photoperiod included 6 replicates with 2 turtles per replicate. All turtles were reared for 28 days under corresponding light and stocking density conditions, and the other conditions were kept the same as the period of acclimation. Removing uneaten food and feces and refreshing water under constant darkness were performed in dim red light with an intensity of 3 lux [9]. Food intake of the intermediate 3 days in each week was recorded for each replicate (one replicate inside one rearing tank).

### 2.3. Metabolic Rates Measurement

After 28 days of treatment, all turtles were fasted for 2 days and weighed. Twelve individuals were measured for the standard metabolic rate (SMR) under corresponding light conditions by random selection from each group. The SMR of each turtle was measured by monitoring changes in oxygen concentrations using an oxygen sensor (FireStingO_2_, Pyro Science, Aachen, Germany) in a cylindrical airtight chamber at 28 °C (diameter = 20 cm, height = 6.5 cm). Each turtle was placed into the chamber to acclimate for 30 min with the cover off, and then the cover was closed tightly, and the oxygen level inside the chamber was recorded for 60 min (at 60 times per second). The final oxygen concentration was not less than 75% of the saturation oxygen concentration, with no influence on the turtles’ respiration.

### 2.4. Sampling

After 5 days of recovery of SMR measurement, all turtles were fasted for 2 days again. Then, three turtles per tank in the high-density groups and one turtle per tank in the low-density groups were sacrificed by fast decapitation during the day from 11:00 a.m. to 13:00 p.m. (about 6 h into daytime in 12L:12D) and at night from 23:00 p.m. to 1:00 a.m. (about 6 h into nighttime in 12L:12D) [36]. Blood was collected with heparinized tubes. The brain was resected quickly on ice and frozen in liquid nitrogen. The blood was centrifuged for 10 min at 4 °C and 3000 r/min, and then the upper pale-yellow plasma was reserved. Finally, tissues and plasma were stored at −80 °C.

### 2.5. Calculations

The specific growth rate (SGR, %/day), food intake (FI, g/ind./day), and SMR (mL/g/h) were calculated according to the following formulas:(1)$$\mathrm{SGR}=100\times \frac{{\mathrm{lnW}}_{f\;}-{\mathrm{lnW}}_{i}}{T}$$
(2)$$\mathrm{FI}=\frac{F_{0\;}-F_{t}}{N}$$
where W_f_ and W_i_ are the average body weights of each replicate at the beginning and the end of the experiment, respectively; T is the duration of treatment (e.g., 28 days); F_0_ and F_t_ are the total weights of feeding diets and the weights of the residual diets each day (dry weight, g), respectively; and N is the number of individuals in each group.
(3)$$\mathrm{SMR}=\frac{\Delta O_{2}V}{\mathrm{tW}}$$
where ∆O_2_ is the change in oxygen level in the respiration chamber; V is the volume of the respiration chamber minus the weight of the measured turtle; t is the respiration time; and W is the body weight of the measured turtle.

### 2.6. Analysis of Plasma Hormones

The melatonin, leptin, ghrelin, growth hormone (GH), triiodothyronine (T_3_), and thyroxine (T_4_) concentrations and T_4_5′-deiodinase activity in the plasma were measured using a double-antibody sandwich-enzyme-linked immunosorbent assay (FANKEL Industrial Co., Ltd., Shanghai, China). All assays were performed according to the manufacturer’s instructions. Briefly, 50 μL of different concentrations of standards and diluted samples were added to standard wells and sample wells in microplates, respectively. Next, the microplate was incubated at 37 °C for 30 min and then cleaned with wash water 5 times. One hundred microliters of horseradish peroxidase (HRP) coupling reagent was added to each well, and the microplate with reagents was incubated at 37 °C for another 30 min. Then, the microplate was cleaned with wash water 5 times, and 50 μL of chromogenic reagent A and 50 μL of chromogenic reagent B were added. Following a 10 min incubation, 50 μL of stop solution was added to each well to stop the reaction. The results were measured within 15 min using a microplate reader equipped with a 450 nm filter. The enzyme activity of samples was determined by the linear regression equation, which was calculated on the basis of standard density and OD value. Each sample was tested in triplicate.

### 2.7. Total RNA Extraction, First-Strand cDNA Synthesis, and Real-Time PCR

Total RNA was extracted from the brain tissues using the traditional TRIzol method [37]. In brief, 30–50 mg of tissue was homogenized in 1 mL of TRIzol reagent (Invitrogen, Carlsbad, CA, USA) using an OMNI Bead Ruptor 24 (OMNI International Homogenizer Company, Kennesaw, GA, USA). The homogenized samples were centrifuged at 4 °C at 12,000× *g* for 10 min, and the upper liquid was transferred to new EP tubes and incubated at room temperature for 5 min to permit the complete dissociation of nucleoprotein complexes. Then, approximately 300 μL of chloroform was added to each EP tube, mixed vigorously, and centrifuged at 12,000× *g* for 15 min at 4 °C. After centrifugation, the upper colorless aqueous phase was transferred into a new EP tube; afterward, 300 μL of isopropanol was added to each EP tube, mixed gently, and incubated for one night at 4 °C. The supernatant was discarded, and the RNA pellet was washed with 1 mL of 75% ethanol. The RNA pellet was air-dried and dissolved in RNase-free water. The quality and quantity of the acquired RNA were assessed by electrophoresis and a NanoDrop 2000 spectrophotometer (Thermo, Waltham, MA, USA), respectively (1.9 < A260/280 < 2.1, c: 0.4–1 μg/μL).

First-strand cDNA was synthesized with 2 μg of total RNA as template using the PrimeScript II 1st strand cDNA synthesis kit (Takara, Japan). The acquired cDNA was stored at −20 °C for quantitative real-time PCR.

The real-time PCR primers were designed using the online software prime3 (http://primer3.ut.ee/, version 4.1.0). All of the primers were verified by sequencing, and real-time PCR primer information is presented in Table 1. Real-time PCR was conducted on a 7500 real-time PCR system (Applied Biosystems, Carlsbad, CA, USA). The reaction system was 20 μL, containing 10 μL of 2× TransStart Top Green qPCR SuperMix (+Dye II) (AQ132-2, TransGen Biotech, Beijing, China), 3 μL of cDNA template, 1 μL of each forward and reverse primer (5 μmol L^−1^), and 5 μL of Milli-Q H_2_O. Glyceraldehyde-3-phosphate dehydrogenase (*GAPDH*) was selected as the internal control gene. Transcript levels of the same gene in different samples were expressed relative to the mean control (12L:12D in low density during day sampling) value that was set to 1.0. The relative gene expression was calculated using the comparative CT method [38].

### 2.8. Statistical Analysis

Data are presented as mean ± standard error. Statistical analyses were conducted with SPSS 17.0. The data were firstly checked for normality with the Kolmogorov–Smirnov test and for homogeneity of variances with Levene’s test. If these assumptions were met, a multivariate analysis of variance (ANOVA) was performed. Then, the effects of photoperiod under each specific density were assessed by one-way ANOVA followed by Duncan’s post-hoc test to compare means among the groups. The effects of density and sampling time were assessed by independent-sample *t*-tests. Otherwise, Kruskal–Wallis test followed by Mann–Whitney U post-hoc test was used. *p* < 0.05 was set as the statistically significant level.

## 3. Results

### 3.1. Plasma Melatonin Level

There was no interaction among photoperiod, sampling time, and density on plasma melatonin level, but each of the three factors showed a significant effect on plasma melatonin independently (Table 2). The melatonin level during daytime was the lowest in 24L:0D under low density, while it was clearly the highest in 0L:24D during nighttime under high density. Plasma melatonin levels sampled at night were all higher than those sampled during the day. Plasma melatonin levels of the high-density groups were significantly higher than those of the low-density groups during daytime and nighttime (Table 2, Figure 1).

### 3.2. Growth and Food Intake

Two-way ANOVA revealed that there was no significant interaction between photoperiod and density in SGR and FI (SGR: F_2,18_ = 0.307, *p* = 0.740; FI: F_2,72_ = 0.988, *p* = 0.377). However, photoperiod showed a significant effect on FI (SGR: F_2,18_ = 0.927, *P* = 0.414; FI: F_2,72_ = 9.996, *p* < 0.001), and density significantly affected the two parameters (SGR: F_1,18_ = 4.478, *p* = 0.049; FI: F_1,72_ = 21.037, *p* < 0.001). Under high density, SGR of the 0L:24D group was significantly higher than that of 24L:0D group, but photoperiod showed no clear effect on SGR under low density (high density: F_2,3_ = 7.149, *p* = 0.072; low density: F_2,15_ = 0.420, *p* = 0.664). SGR of the high-density group was significantly lower than that of the low-density group (t_16.670_ = −2.952, *p* = 0.009) (Figure 2A). FI of the 0L:24D group was significantly higher than those of the 24L:0D and 12L:12D groups (high density: F_2,36_ = 6.088, *p* = 0.005; low density: F_2,36_ = 5.031, *p* = 0.012). FI in the high-density groups was significantly lower than those of low-density groups (t_76_ = −4.125, *p* < 0.001) (Figure 2B).

### 3.3. Appetite Regulation-Related Gene Expressions and Plasma Leptin and Ghrelin Levels

Photoperiod, sampling time, and density did not significantly interact with the expression levels of the appetite regulation-related *Kiss1*, *CART*, and *NPY* genes, while a significant interaction between photoperiod and time was present in the expression of *NPY* (Table 2). The expression of the *Kiss1* gene in 24L:0D was the highest among the three photoperiod treatments under low density during the day. *CART* gene expression remained stable in all of the treatment groups. During the day sampling time, *NPY* expression in 12L:12D was significantly higher than that in 24L:0D under high density, and all *NPY* expression levels were higher in the high-density groups than those in the low-density treatment (Table 2, Figure 3A–C).

Multivariate ANOVA results showed that there were no interactions among photoperiod, time, and density on plasma leptin and ghrelin, while there was a significant interaction between photoperiod and time (Table 2). The leptin level in 0L:24D under low density was significantly higher at night than at daytime. Ghrelin levels in 0L:24D under high density, 24L:0D, and 0L:24D under low density were also significantly higher at night than at daytime (Table 2, Figure 3D,E).

### 3.4. SMR

Two-way ANOVA results showed that there was no significant interaction between photoperiod and density on SMR (F_2,66_ = 0.269, *p* = 0.765), while photoperiod displayed a significant effect on SMR (F_2,66_ = 4.936, *p* = 0.010). Despite the lack of a significant difference, SMR showed an increasing trend as the dark phase increased (high density: F_2,33_ = 2.249, *p* = 0.121; low density: F_2,33_ = 3.031, *p* = 0.062, Figure 4). Furthermore, when density groups were combined, SMR in 24L:0D was significantly lower than that in 0L:24D (F_2,69_ = 5.061, *p* = 0.009, Figure 4).

### 3.5. Plasma Growth Hormone Level

There were no interactions among photoperiod, time and density on plasma growth hormone GH contents, while sampling time showed a significant influence on GH (Table 2). GH in 0L:24D under low density was significantly higher at night than that at daytime (Figure 5).

### 3.6. Thyroid Activity

There was a significant interaction among photoperiod, time, and density on the T_3_ level but no interaction on T_4_ and T_4_5′-deiodinase levels. All three factors significantly affected plasma T_4_ levels, photoperiod and density showed clear effects on T_4_5′-deiodinase activity (Table 2). Plasma T_3_ level in 0L:24D under high density during the day was the highest among the three photoperiod groups, and that of 24L:0D was the lowest, but under low density, it was higher at night than during the day. Plasma T_4_ level in 12L:12D under high density during the day was significantly higher than that in 24L:0D. Moreover, T_4_ levels in 24L:0D and 0L:24D were higher at night than during the day. Both T_3_ and T_4_ levels under high density during the day were significantly higher than those under low density. Plasma T_4_5′-deiodinase activities in 12L:12D under low density at night were significantly higher than that in 24L:0D, and those under high density were significantly higher than those under low density, whether during the day or at night (Figure 6).

## 4. Discussion

This study investigated how extreme photoperiods and stocking densities affected the melatonin, growth, and energy metabolism of the soft-shelled turtle at the individual, physiological, and molecular levels. We found that plasma melatonin concentrations were the highest in 0L:24D and the lowest in 24L:0D, and the level in 12L:12D at night was also significantly higher than that during the day (Figure 1). These results indicate that the turtle perceived light changes and darkness promoted melatonin production. Kumari et al. [1] also found that in some mammals, serum melatonin levels under constant light were significantly lower than those under constant darkness and 12L:12D. It is known that melatonin is mainly synthesized during the dark phase [39,40,41], thus matching our results. Interestingly, plasma melatonin levels sampled at night were clearly higher than those sampled during the day either under constant darkness or constant light, suggesting that melatonin secretion of the soft-shelled turtle was also likely to be regulated by some unknown endogenous factors. One explanation is possibly endogenous circadian rhythm. Similar results are also revealed in domestic turkey (*Meleagris gallopavo gallopavo var. domesticus*) with higher melatonin concentrations during nighttime than during daytime after 7 days of continuous dim red light treatment [9], which was regarded as endogenously generated diurnal rhythm. Strangely, the circadian rhythm of the sleepy lizard (*Tiliqua rugosa*) was eliminated by 3 days of constant light [10], while higher plasma melatonin concentrations at nighttime remained after a month of constant light treatment in our present study. The difference may be because the Chinese soft-shelled turtle is a benthic living animal and relatively insensitive to light due to its high tolerance to extreme environments, thus mild constant light with 900 lux did not abolish the difference between nighttime and daytime. It is partially supported by the fact that the ratio of melatonin concentrations (night/day) of the soft-shelled turtle in 12L:12D (approximately 1.3-fold) is significantly lower than those of the sleepy lizard (approximately 2.5-fold) [10], rats (more than 10-fold) [36], and birds (2–18-fold) [9]. Another explanation might be that turtles still received external time cues under constant darkness and constant light by the operations of feeding, removing uneaten food and feces, and refreshing water. In addition, lacking multiple measurements of melatonin levels during the 24 h interval caused, we cannot determine the free-running circadian period of turtles under constant darkness and constant light. Thus, whether our results on melatonin secretion under extreme light cycles are also due to endogenous circadian rhythm needs further study. High stocking density stress augmented plasma melatonin concentrations of the turtle (Figure 1). This result conflicts with the reports of fish, in which plasma melatonin decreased in rainbow trout (*Oncorhynchus mykiss*) [42] and unchanged in tilapia (*Oreochromis mossambicus*) [43] under high stocking density. The different responses of melatonin to high stocking density might be species-specific, or because of differences in means of treatment, as the fish were treated by acute, short-period stress, while the turtles were exposed to chronic stress in the present study.

The SGR showed an increasing trend as the dark period increased under high densities, and food intake was the highest in 0L:24D among the three photoperiod treatments. Similar results were also revealed in the soft-shelled turtle (*Lissemys punctata punctata*), which consumed more food and increased body weight when exposed to a short photoperiod [44]. High stocking density (38.10 ind./m^2^) showed marked adverse effects on the growth of the soft-shelled turtle juveniles. The specific growth rate (SGR) and food intake under high density were significantly lower than those under low density (Figure 2). This is in agreement with our previous study that the SGR under 3.13 ind./m^2^ was higher than the SGR under 18.75 ind./m^2^ in juvenile soft-shelled turtles [35]. These results suggested that constant darkness can improve growth by increasing food intake and alleviating the adverse effects of high stocking density at least in part in the soft-shelled turtle juveniles.

Empirical studies have revealed that there is a close relationship between melatonin and appetite-modulating hormones [7,14,31,45]. *Kiss1* and *CART* genes are associated with the regulation of energy homeostasis by increasing energy expenditure and inhibiting feeding [46,47]. In the present study, during daytime in 24L:0D, *Kiss1* mRNA increased under low density, while *NPY* mRNA decreased under high density, suggesting that lower SGR and food intake under constant light may be due to upregulation of *Kiss1* and downregulation of *NPY* genes. Unexpectedly, the plasma leptin and ghrelin contents kept stable in all treatments, although constant darkness elevated the plasma melatonin level (Figure 1). One possible explanation is that the melatonin effects may be dose dependent. Because most reported studies on melatonin effects were conducted by externally imported melatonin with a higher dose than the endogenous melatonin levels, they found that melatonin elicits a decrease in body mass and/or food consumption by increasing leptin and reducing ghrelin levels, while a short photoperiod induces increases in body mass and food intake by decreasing leptin levels or increasing ghrelin levels [7,14,31,45].

The SMR showed an evident increasing tendency as the dark phase increased, and SMR in 0L:24D was significantly higher than that in 24L:0D when combining high density with low density. A higher SMR in constant darkness is likely correlated with a higher growth rate and food intake (Figure 2). Plasma GH contents remained mostly unchanged except for the GH level in constant darkness, which was higher at night than during the day. Ghrelin is a ligand for the growth hormone secretagogue receptor, regulating the release of growth hormone from the pituitary [48]. Therefore, unchanged GH contents are possibly associated with stable plasma ghrelin levels (Figure 3). Thyroid hormones play important roles in the regulation of growth, development, and energy homeostasis. The reported responses of thyroid activity to photoperiod varied with species [44,49,50,51]. In the present study, the thyroid hormones and enzymatic activities mainly remained stable in different photoperiods, while T_4_ in 24L:0D under high density during the day and T_4_5′-deiodinase activity in 24L:0D under low density at night were significantly reduced compared with those of the control, and T_3_ under high density during the day increased with increasing dark phase. These results may suggest that constant light inhibits while constant darkness stimulates thyroid activity. Similar results also revealed that a short photoperiod activated thyroid activity with increasing relative thyroid weight and peroxidase activity, while a long photoperiod suppressed thyroid activity in soft-shelled turtles (*Lissemys punctata punctata*) [44]. Creighton and Rudeen [52] found that the administration of melatonin increased serum T_3_ level in *Mesocricetus auratus*. Moreover, it has been reported that the administration of T_3_ in the brain by injection upregulated *NPY* and *AgRP* mRNAs and downregulated *POMC* mRNA and thus increased food intake [53,54]. Therefore, the effects of photoperiods on growth and food intake in soft-shelled turtles are likely involved in the process of constant light inhibition, while constant darkness stimulates the secretion of melatonin, which acts on the thyroid and stimulates the production of T_3_, eliciting the decrease and increase in food intake and metabolism in constant light and darkness, respectively. Finally, the gained energy by feeding exceeded expenditure of metabolism, exhibiting higher growth in constant darkness. In addition, the T_4_ contents in 24L:0D and 0L:24D at night were higher than those during the day in both densities, and T_3_ level in 0L:24D under low density at night was also higher than that during the day, which further suggests that the increase in thyroid activity is likely associated with elevated melatonin levels. High stocking density increased thyroid activity with higher plasma T_4_, T_3_, and T_4_5′-deiodinase activity when compared to low density, suggesting elevated energy expenditure and metabolic levels under high density. Although high stocking density also contributed to the production of melatonin and increase of thyroid hormones, the food intake decreased with stocking density, which is possibly related to the limitation of activity area and thus the decrease in food availability. Thus, the higher metabolism and lower food intake caused the decrease in growth under high stocking density.

## 5. Conclusions

Melatonin secretion of the soft-shelled turtle was not only affected by light, but also likely to be regulated by unknown endogenous factors and density. Constant darkness can improve growth by increasing food intake and alleviating the adverse effects of high stocking density in Chinese soft-shelled turtle juveniles, while constant light decreased appetite and metabolism, thus inhibiting growth. Furthermore, it seems that the altered melatonin concentrations induced by constant darkness and high density affect growth and food intake by modulating energy metabolism (elevated thyroid activity and SMR) rather than appetite (unchanged leptin and ghrelin contents, *Kiss1*, *CART*, and *NPY* mRNA) in Chinese soft-shelled turtles.

## Figures and Tables

**Figure 1 biology-11-00965-f001:**
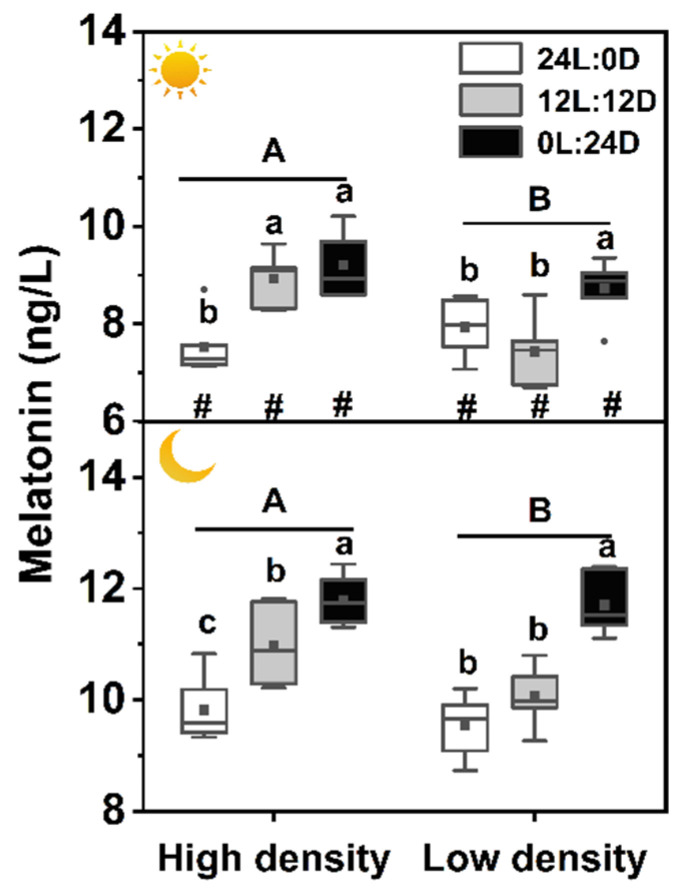
Effects of photoperiod and density on plasma melatonin levels in *Pelodiscus sinensis*. 
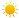
 and 
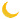
 indicate daytime and nighttime, respectively. Data are *n* = 5 or 6 for all groups. Data are presented as the mean ± S.E. In the box plots, the solid horizontal indicates median, the filled square indicates the mean, and points indicate outliers. Different lowercase letters denote statistically significant differences among different photoperiods in each density; different capital letters indicate a significant difference between different densities; # indicate a significant difference between the daytime and nighttime (*p* < 0.05).

**Figure 2 biology-11-00965-f002:**
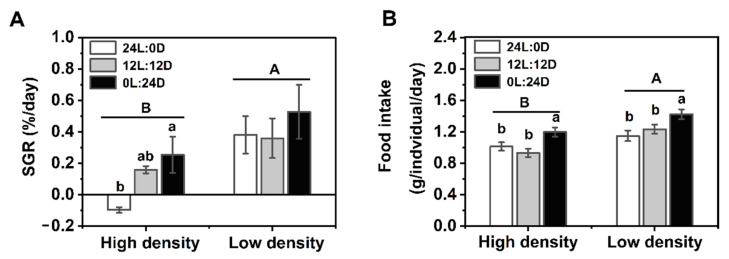
Effects of density and photoperiod on SGR (**A**) and food intake (**B**) of *Pelodiscus sinensis*. Data are *n* = 24 under high density and *n* = 12 under low density for SGR, *n* = 13 for all groups in food intake. Data are presented as the mean ± S.E. Different lowercase letters denote a statistically significant difference among different photoperiods under each density, and different capital letters indicate a significant difference between different densities (*p* < 0.05).

**Figure 3 biology-11-00965-f003:**
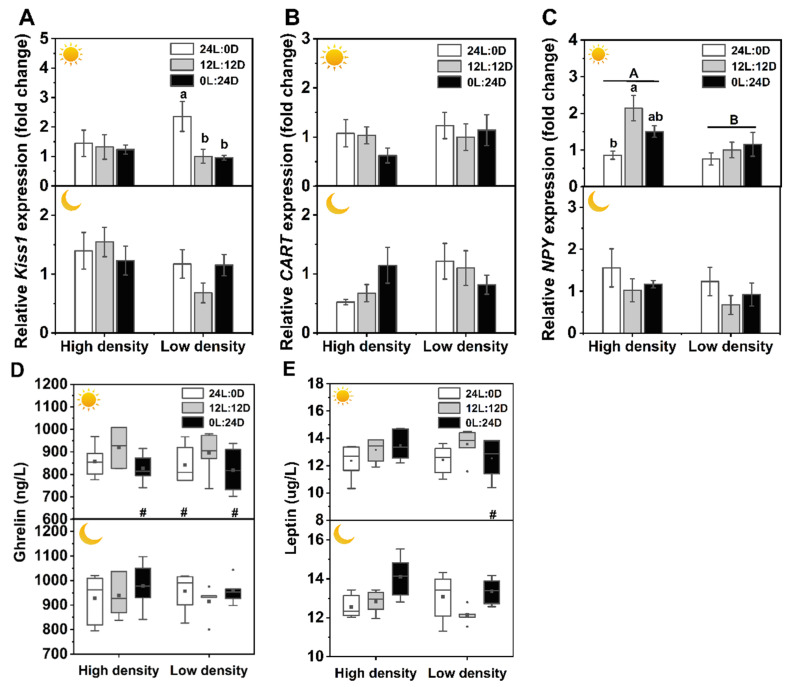
Effects of photoperiod and density on the mRNA expression levels of appetite-related genes in the brain, plasma leptin and ghrelin in *Pelodiscus sinensis*. *Kiss1* (**A**), *CART* (**B**), *NPY* (**C**), plasma leptin (**D**), and plasma ghrelin (**E**). 
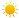
 and 
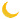
 indicate daytime and nighttime, respectively. Dates are *n* = 5 or 6 for all groups. Data are presented as the mean ± S.E. In (**A**)–(**C**), different lowercase letters denote statistically significant differences among different photoperiods in each density; different capital letters indicate a significant difference between different densities (*p* < 0.05). In (**D**,**E**), the description of the box plots is the same as that shown in Figure 1. # indicate a significant difference between the daytime and nighttime (*p* < 0.05).

**Figure 4 biology-11-00965-f004:**
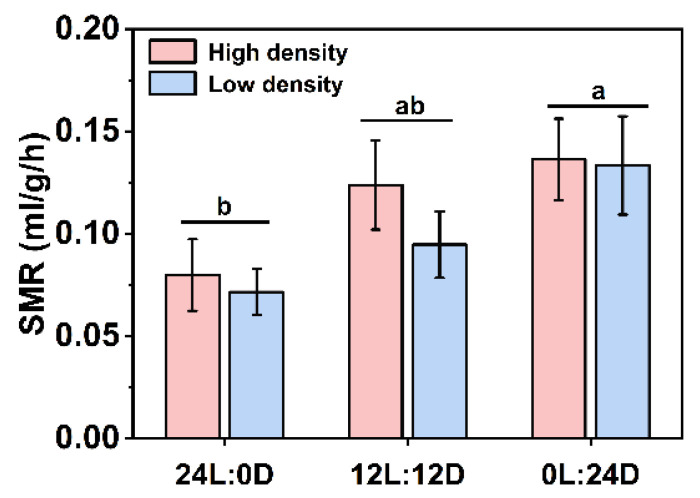
Effects of photoperiod and density on the SMR of *Pelodiscus sinensis*. Data are *n* = 12 for all groups. Data are presented as mean ± S.E.; different lowercase letters indicate a significant difference among different photoperiods (*p* < 0.05).

**Figure 5 biology-11-00965-f005:**
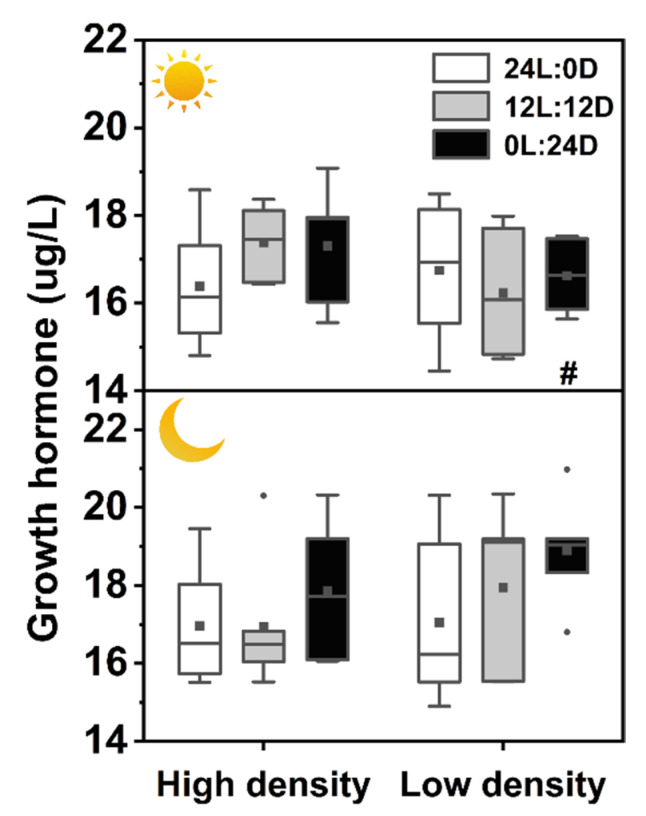
Effects of photoperiod and density on plasma growth hormone of *Pelodiscus sinensis*. 
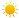
 and 
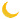
 indicate daytime and nighttime, respectively. Data are *n* = 5 or 6 for all groups. The description of the box plots is the same as that shown in Figure 1. # indicate a significant difference between the daytime and nighttime (*p* < 0.05).

**Figure 6 biology-11-00965-f006:**
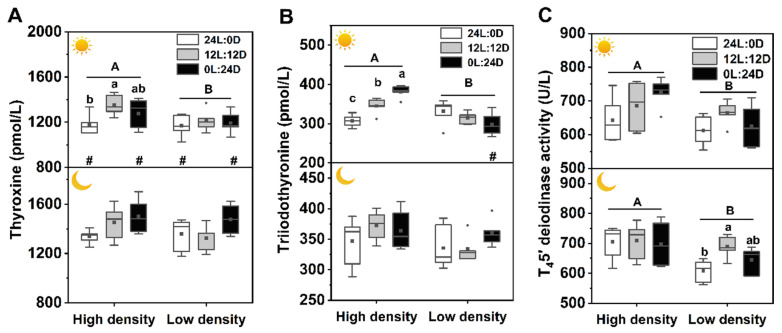
Effects of photoperiod and density on plasma T_4_ (**A**), T_3_ (**B**), and T_4_5′-deiodinase activity (**C**) of *Pelodiscus sinensis*. 
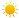
 and 
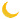
 indicate daytime and nighttime, respectively. Data are *n* = 5 or 6 for all groups. Data are presented as the mean ± S.E. The description of the box plots is the same as that shown in Figure 1. Different lowercase letters denote statistically significant differences among different photoperiods under each density; different capital letters indicate a significant difference between different densities; # indicate a significant difference between the daytime and nighttime (*p* < 0.05).

**Table 1 biology-11-00965-t001:** Gene name, GenBank accession number, forward and reverse primer sequences, and real-time PCR efficiency for target genes analyzed.

Gene Name	Accession No.	Forward Primer	Reverse Primer	PCR Efficiency
*GAPDH*	NM_001286927.1	TTCATGGCACTGTCAAGGCT	GGTTGACGCCCATCACAAAC	97.1%
*Kiss1*	XM_025189675.1	GAGTTCCAGTTGTAGGCGGA	GGGAGCCTGCAGATGGATAT	90.7%
*CART*	XM_025185227.1	CAAGCGGATCCCCATTTACG	TTCCCGATTCGAGCTCCTTT	89.0%
*NPY*	XM_006138369.3	TCATCACGCGGCAGAGATAT	GTCTTCAAACCTAGATCTTGGGA	97.9%

**Table 2 biology-11-00965-t002:** Multivariate analysis of variance of photoperiod, time and density on plasma melatonin, food intake and energetic metabolism genes and hormones.

Gene	Photoperiod	Time	Density	Density × Time	Photoperiod × Density	Photoperiod × Time	Density × Photoperiod × Time
Melatonin	F_2,58_ = 45.800	F_1,58_ = 274.608	F_1,58_ = 11.037	F_1,58_ = 0.106	F_2,58_ = 7.027	F_2,58_ = 2.881	F_2,58_ = 1.967
	***p* < 0.001**	***p* < 0.001**	***p* = 0.002**	*p* = 0.746	***p* = 0.002**	*p* = 0.064	*p* = 0.149
*Kiss1*	F_2,57_= 2.440	F_1,57_ = 1.319	F_1,57_ = 0.461	F_1,57_ = 1.583	F_2,57_ = 2.555	F_2,57_ = 1.727	F_2,57_ = 0.961
	*p* = 0.096	*p* = 0.256	*p* = 0.500	*p* = 0.213	*p* = 0.087	*p* = 0.187	*p* = 0.388
*CART*	F_2,53_ = 0.115	F_1,58_ = 0.534	F_1,53_ = 2.806	F_1,53_ = 0.031	F_2,53_ = 0.475	F_2,53_ = 0.623	F_2,53_ = 2.485
	*p* = 0.892	*p* = 0.468	*p* = 0.100	*p* = 0.862	*p* = 0.625	*p* = 0.540	*p* = 0.093
*NPY*	F_2,53_ = 0.620	F_1,53_ = 1.776	F_1,53_ = 6.945	F_1,53_ = 1.230	F_2,53_ = 1.982	F_2,53_ = 5.911	F_2,53_ = 0.719
	*p* = 0.542	*p* = 0.188	***p* = 0.011**	*p* = 0.272	*p* = 0.148	***p* = 0.005**	*p* = 0.492
Leptin	F_2,58_ = 3.504	F_1,58_ = 0.157	F_1,58_ = 0.933	F_1,58_ = 0.099	F_2,58_ = 2.001	F_2,58_ = 4.180	F_2,58_ = 1.064
	***p* = 0.037**	*p* = 0.694	*p* = 0.338	*p* = 0.755	*p* = 0.144	***p* = 0.020**	*p* = 0.352
Ghrelin	F_2,58_ = 0.552	F_1,58_ = 19.367	F_1,58_ = 0.323	F_1,58_ = 0.080	F_2,58_ = 0.229	F_2,58_ = 3.439	F_2,58_ = 0.207
	*p* = 0.579	***p* < 0.001**	*p* = 0.572	*p* = 0.778	*p* = 0.796	***p* = 0.039**	*p* = 0.814
GH	F_2,58_ = 1.858	F_1,58_ = 4.712	F_1,58_ = 0.085	F_1,58_ = 2.467	F_2,58_ = 0.061	F_2,58_ = 0.603	F_2,58_ = 0.977
	*p* = 0.165	***p* = 0.034**	*p* = 0.771	*p* = 0.122	*p* = 0.941	*p* = 0.551	*p* = 0.382
T_4_	F_2,58_ = 5.769	F_1,58_ = 48.356	F_1,58_ = 5.553	F_1,58_ = 0.444	F_2,58_ = 2.597	F_2,58_ = 2.716	F_2,58_ = 0.090
	***p* = 0.005**	***p* < 0.001**	***p* = 0.022**	*p* = 0.508	*p* = 0.083	*p* = 0.075	*p* = 0.914
T_3_	F_2,58_ = 3.706	F_1,58_ = 12.832	F_1,58_ = 14.999	F_1,58_ = 0.916	F_2,58_ = 6.479	F_2,58_ = 0.006	F_2,58_ = 7.830
	***p* = 0.031**	***p* = 0.001**	***p* < 0.001**	*p* = 0.342	***p* = 0.003**	*p* = 0.994	***p* = 0.001**
T_4_5′-deiodinase	F_2,58_ = 4.303	F_1,58_ = 1.617	F_1,58_ = 17.497	F_1,58_ = 0.065	F_2,58_ = 1.703	F_2,58_ = 0.607	F_2,58_ = 1.660
	***p* = 0.018**	*p* = 0.209	***p* < 0.001**	*p* = 0.799	*p* = 0.191	*p* = 0.548	*p* = 0.199

*p*-Values in bold indicate statistically significant values.

## Data Availability

The data presented in this study are available on request from the corresponding author.

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
