# Peer review of "Effects of Extreme Light Cycle and Density on Melatonin, Appetite, and Energy Metabolism of the Soft-Shelled Turtle (Pelodiscus sinensis)"

_biology, 2022, doi:10.3390/biology11070965_

Round 1

Reviewer 1 Report

the manuscript entitle: Effects of extreme light cycle and density on melatonin, appetite and energy metabolism of the soft-shelled turtle (Pelodiscus sinensi) describes the influence of light on many biochemical parameters and on appetite. the study need to be improved with a major revision before it re-evaluation. 

define extreme light cicle.

lines 17-19 should be rewritten, this assetment is not correct

lines19-21 the sentence is not clear for a reader that does not know the main text

in the abstract section the material and method describtion should be rewritten. it is not clear which parameters were investigated, when the blood collection were performed, which statistical analysis were applied. conclusion is not clear, should be rewritten.

introduction

line 77 also reproduction is influenced by melatonin (Chronobiol int 2020, 37(7) page 974-979). different photoperiod had an influence on the serum melatonin oscillation (Chronobiol int 2018, 35 (3), page 329-335).

many species lose the circadian oscillation of serum melatonin in constant light conditions (Bio, Rhyth Res, 2013, 44 (1), page 143-149) should it be the same for turtle?

clarify the aim of the study

2.2 experimental protocol

clarify the group constitution.)

how was the time of the data collection chosen?

the authors can't talk about circadian rhythm, the experimental protocol and the statistical analysis applied are not appropriated for study on circadian evaluation.

What the three way anova is?

conclusion should be rewritten deleting the reference to circadian rhythm  

Reviewer 2 Report

Dear Authors

Many thanks for your well designed study

Please just check for spelling and grammar errors.

Author Response

Response to Reviewer 2 Comments

Point 1: Please just check for spelling and grammar errors.

Response 1: Thanks for your suggestion. We have checked for spelling and grammar errors carefully.

Reviewer 3 Report

The manuscript presents the results of examination of the effect of constant light and constant darkness on melatonin secretion and melatonin modulated appetite and energy metabolism in the soft-shelled turtle. The authors hypothesized that “increased melatonin levels induced by darkness would increase appetite and energy metabolism, and thus promote growth…”, they measured twice a day melatonin and some other hormones (plus several other biological indexes), and they concluded that “altered plasma melatonin induced by constant darkness and density seems to be involved in the modulation of energy metabolism rather than appetite in the soft-shelled turtle.”

The major problem of the study is that there is no reliable information of melatonin levels because there were no multiple measurements of melatonin during a 24-h interval of the day after 28 days spent in the constant lighting conditions. (“Three turtles per tank in the high-density groups and one turtle per tank in the low density groups were sacrificed by fast decapitation during the day at 11:00 am -13:00 pm and at night at 23:00 pm-1:00 am.”). Therefore, “subjective night “ and “subjective day” intervals remained unknown, while, due to a shorter or a longer free-running circadian period, 11:00 am -13:00 pm might be already the subjective night hours while 23:00 pm-1:00 am might be the subjective day hours (at the 29th day). Another explanation might be that the animals still received information about the time of the day due to their incomplete isolation from the external time cues during 28 days. At baseline, “Uneaten food and feces were removed after one hour of feeding, and approximately half of the water was refreshed in the tank each day.” If something like that happened during the 28-day experiment, the circadian rhythms can be entrained by such kinds of social signals (additionally accompanied by light intervention during the procedure because people must recognize these “food and feces”).

Abstract

All abbreviation (e.g., standard metabolic rate (SMR)) must be explained in Abstract. On the other hand, the abbreviation (e.g., SGR) is not necessary in Abstract when specific growth rate was mentioned only once.

“Together, melatonin secretion is regulated by endogenous rhythm and density other than light.” Must be rewritten.

Introduction

“on physiological progress in turtles.” Processes?

“Weather turtles have endogenous circadian rhythm of melatonin secretion are unclear yet.” Pls briefly review the studies on the circadian rhythm of melatonin in turtles (e.g., Skene DJ, Vivien-Roels B, Pevet P. Pineal 5-methoxytryptophol rhythms in the box turtle: effect of photoperiod and environmental temperature. Neurosci Lett. 1989 Mar 13;98(1):69-73. doi: 10.1016/0304-3940(89)90375-3. PMID: 2710400).

“we speculate that the better growth of Chinese soft-shelled turtle in constant darkness might be involved in the increased secretion of melatonin.” Must be rewritten.

“the circadian fluctuations of plasma melatonin under extreme light cycle (constant light and constant darkness); “ The twice a day measurement does not provide reliable information on the circadian rhythm (must be mentioned as limitation in the limitation paragraph of Discussion).

“The soft-shelled turtle…” This and the following sentences must be moved up in introduction section.

Discussion

Limitation paragraph must be included prior Conclusions.

Conclusions

“In conclusion,” must be excluded.

“melatonin secretion of the soft-shelled turtle was not only affected by light, but also regulated by endogenous rhythm and density.” No evidence was provided – see the comment on the major problem of this study.

“Furthermore, it seems that the altered melatonin concentrations induced by constant darkness and density affect growth and food intake by modulating energy metabolism rather than appetite in Chinese soft-shelled turtles with unchanged anorexigenic signals (leptin contents, Kiss1 and CART mRNA) and orexigenic signals (ghrelin contents, NPY mRNA) in different photoperiods and densities, but elevated thyroid activity in constant darkness and under high stocking density.” Too long sentence. Pls rewrite.

Round 2

Reviewer 3 Report

Although the authors tried to revise the manuscript, they have chosen a wrong way to deal with the effect of circadian timing on their results. In their reply, they written: "We indeed did not examine any circadian
rhythm of melatonin level, as this is not our study purpose. Very sorry for our improper description.We have deleted the background about circadian oscillation in introduction and revised the discussion and conclusion." However, their ignorance in respect of the circadian science led them to even worse statements. For instance, they added:

"In addition, a day of 24 h is also divided into 2 period for constant darkness and constant light treatment groups, that is, the “subjective day” and “subjective night”. The “subjective day” means the period of the photophase of 12L:12D, that is, 6:00-18:00; likewise, the “subjective night” means the period of the scotophase of 12L:12D, that is, 18:00-
6:00. In order to understand easily for readers, we have added the information, please see page 3, line 136-138."

I recommend to read the literature on the circadian rhythm to learn what is " subjective day" and "subjective night" to avid their misleading usage.

Round 3

Reviewer 3 Report

As I mentioned in my first review, “the major problem of the study is that there is no reliable information of melatonin levels because there were no multiple measurements of melatonin during a 24-h interval of the day after 28 days spent in the constant lighting conditions. (“Three turtles per tank in the high-density groups and one turtle per tank in the low density groups were sacrificed by fast decapitation during the day at 11:00 am -13:00 pm and at night at 23:00 pm-1:00 am.”). Therefore, “subjective night “ and “subjective day” intervals remained unknown, while, due to a shorter or a longer free-running circadian period, 11:00 am -13:00 pm might be already the subjective night hours while 23:00 pm-1:00 am might be the subjective day hours (at the 29th day). Another explanation might be that the animals still received information about the time of the day due to their incomplete isolation from the external time cues during 28 days. At baseline, “Uneaten food and feces were removed after one hour of feeding, and approximately half of the water was refreshed in the tank each day.” If something like that happened during the 28-day experiment, the circadian rhythms can be entrained by such kinds of social signals (additionally accompanied by light intervention during the procedure because people must recognize these “food and feces”).”

The authors cannot improve their study by the simple changing the terms as “subjective night “ and “subjective day” on the terms “daytime” and “nighttime”. I can suggest they discuss this methodological issue in more details by adding a limitation paragraph to the end of Discussion.
